# CONTRASTIVE AND MULTI-TASK LEARNING ON NOISY BRAIN SIGNALS WITH NONLINEAR DYNAMICAL SIGNATURES

## ABSTRACT

We introduce a two-stage multitask learning framework for analyzing Electroencephalography (EEG) signals that integrates denoising, dynamical modeling, and representation learning. In the first stage, a denoising autoencoder is trained to suppress artifacts and stabilize temporal dynamics, providing robust signal representations. In the second stage, a multitask architecture processes these denoised signals to achieve three objectives: motor imagery classification, chaotic versus non-chaotic regime discrimination using Lyapunov exponent-based labels, and self-supervised contrastive representation learning with NT-Xent loss. A convolutional backbone combined with a Transformer encoder captures spatial-temporal structure, while the dynamical task encourages sensitivity to nonlinear brain dynamics. This staged design mitigates interference between reconstruction and discriminative goals, improves stability across datasets, and supports reproducible training by clearly separating noise reduction from higher-level feature learning. Empirical studies show that our framework not only enhances robustness and generalization but also surpasses strong baselines and recent state-of-the-art methods in EEG decoding, highlighting the effectiveness of combining denoising, dynamical features, and self-supervised learning.

## 1 INTRODUCTION

Electroencephalography (EEG) signals are widely used to study human brain activity in both clinical and cognitive neuroscience applications. One of the key challenges in EEG-based Brain-Computer Interfaces (BCIs) is accurately classifying Motor Imagery (MI) tasks by analyzing noisy, non-stationary, and temporally complex neural signals to distinguish between real and imagined motor actions. Furthermore, characterizing the underlying dynamical behavior of EEG signals – whether they exhibit a chaotic attractor or a non-chaotic pattern (periodic, quasiperiodic, or no attractor) – can offer novel insights into the variability of the brain state and the complexity of the signal.

In this work, we explore a multitask learning framework that jointly trains a neural network to perform both MI classification and chaotic/non-chaotic signal identification, while also benefiting from a self-supervised contrastive learning objective. By simultaneously learning to (1) classify EEG signals as real versus imagery motor actions, (2) determine whether the signal dynamics are chaotic or non-chaotic (via Lyapunov exponents estimation), and (3) maximize similarity between augmented views of the same input using contrastive loss, we aim to improve the generalization and robustness of EEG decoding under noisy regimes. Our proposed architecture leverages a convolutional-Transformer backbone, which first encodes raw multi-channel EEG signals into temporally contextualized embeddings. These embeddings are then fed into multiple task-specific heads: a classification head for MI tasks, a chaos detection head trained using ground-truth Lyapunov-based labels, and a projector head for contrastive representation learning using the NT-Xent loss. This design enables the model to learn shared representations that are discriminative, dynamic-aware, and invariant to noise or channel-level perturbations. An overview of our proposed multitask learning architecture is illustrated in Figure 1, depicting the shared encoder and the three task-specific heads. This approach is particularly valuable for applications involving noisy, real-world EEG data, where traditional single-task classifiers may struggle. The integration of dynamical systems theory into a modern deep learning

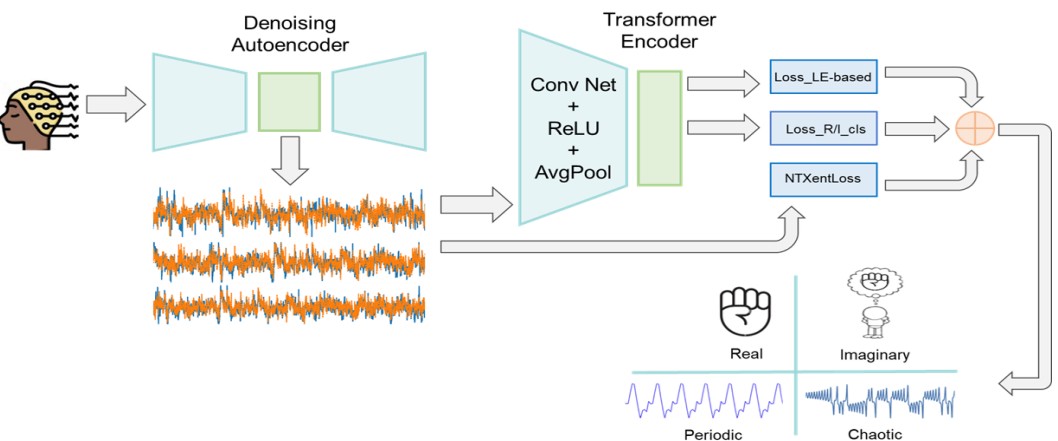

**Figure 1:** Schematic illustration of our multitask learning framework architecture for noisy EEG brain signals.

pipeline contributes a novel mathematical dimension to EEG signal understanding and may offer more interpretable features for downstream clinical or cognitive tasks.

## 2 RELATED WORK

**EEG-based MI classification.** MI classification using EEG has been a central focus in BCI research. Traditional methods include Common Spatial Pattern (CSP) filtering and bandpower feature extraction combined with traditional classifiers such as Linear Discriminant Analysis (LDA) and Support Vector Machines (SVMs) in controlled settings [1, 2]. In recent years, deep learning models, particularly Convolutional Neural Networks (CNNs) like EEGNet [3] and DeepConvNet [4], as well as hybrid architectures combining CNNs with Recurrent Neural Networks (RNNs) or Transformers [5, 6], have demonstrated improved performance by automatically learning hierarchical spatio-temporal features from raw EEG data. Despite these advances, deep models are sensitive to noise, inter-subject variability, and often require extensive preprocessing or calibration efforts [7].

**Contrastive learning in EEG.** Contrastive learning is a powerful self-supervised learning paradigm that allows the extraction of robust and invariant representations without labeled data. Frameworks such as SimCLR [8] and MoCo [9] and their variants have been adapted to physiological signals, including EEG, to learn feature embeddings that remain consistent under noise and data augmentation using time-domain transformations [10, 11, 12, 13]. Recent work indicates that contrastive pretraining improves downstream EEG classification tasks, especially in low-data settings [14]. However, existing methods typically target single downstream tasks, ignoring dynamics like chaos or periodicity.

**Nonlinear dynamical characterization of EEG signals.** EEG signals often exhibit nonlinear dynamics, including chaotic, periodic, and quasiperiodic behaviors, which can be characterized using tools from dynamical systems theory and nonlinear time series analysis such as Lyapunov exponents, correlation dimension, and entropy-based measures [15, 16, 17, 18]. Prior studies have utilized these dynamical features for diagnosing neurological disorders such as epilepsy and Alzheimer's disease [16, 19, 20]. However, incorporating such dynamical characterizations into deep learning models – either as auxiliary supervised/unsupervised tasks or as forms of regularization – is a novel direction. By classifying EEG signals based on their underlying dynamical regimes, the model can be guided to learn representations that better capture the underlying dynamics of neural activity.

**Multitask learning with EEG.** Multitask learning has been applied to EEG data to jointly model related outputs – such as emotion classification, mental workload estimation, or seizure detection – to enhance performance and generalization. For example, Yin et al. [21] proposed a shared representation for emotion classification and workload prediction. Similarly, Lan et al. [22] used multitask learning to jointly predict driver drowsiness and attention from EEG. Multitask learning has also been employed for seizure prediction and detection tasks using shared temporal-spatial features [23]. By sharing the feature extraction layers and incorporating task-specific heads, multitask learning

helps regularize the model and leverage inductive bias across tasks [24, 25, 26]. Other studies have integrated clinical and behavioral outputs [27]. While multitask learning has shown effectiveness across related tasks, its application to jointly learning MI classification [28] and *dynamical signal characterization*—such as identifying underlying behaviors through invariants like the maximal Lyapunov exponent—remains largely unexplored.

**Transformers for EEG.** The Transformer architecture, originally designed for Natural Language Processing (NLP) [29], has recently been applied to time series data, including EEG [30, 31, 32, 33]. Variants such as the Temporal Transformer or Performer have demonstrated the ability to model long-range dependencies [34, 35, 36] and achieve superior performance over RNNs and CNNs in several EEG decoding tasks. Models like TS-TCC [37] and SleepTransformer [38] show that attention-based mechanisms outperform CNNs and RNNs in time-series classification tasks. However, their potential in multitask settings, especially when *combined with contrastive learning and dynamical system labels*, remains under-investigated.

## 3 MATHEMATICAL PRELIMINARIES

A discrete-time Dynamical System (DS) describes the evolution of a state variable over successive time steps according to a deterministic update rule. In many practical settings – particularly in data-driven modeling – DS are extended to include external inputs, resulting in the general formulation

$$z_t = F_{\boldsymbol{\theta}}(z_{t-1}, s_t), \tag{1}$$

where $z_t \in \mathbb{R}^M$ denotes the state of the system at time $t$, $s_t \in \mathbb{R}^N$ is an external control or input signal, and $F_{\boldsymbol{\theta}}$ is a potentially nonlinear function parameterized by a set $\boldsymbol{\theta}$.

A key quantity in the analysis of DS is the *Jacobian matrix*, which captures the local sensitivity of the current state to perturbations in the previous state

$$\boldsymbol{J}_t := \frac{\partial F_{\boldsymbol{\theta}}(z_{t-1}, s_t)}{\partial z_{t-1}} = \frac{\partial z_t}{\partial z_{t-1}}. \tag{2}$$

RNNs represent a parameterized subclass of input-driven DS, in which the state $z_t$ is interpreted as a latent or hidden representation. Architectures such as LSTMs, GRUs, and piecewise-linear RNNs instantiate specific forms of the transition function $F_{\boldsymbol{\theta}}$, enabling them to model complex temporal dependencies in sequential data.

Given $z_1 \in \mathbb{R}^M$ and a sequence of inputs $\boldsymbol{S} = \{s_t\}$, the system (1) evolves recursively as

$$z_T = F_{\boldsymbol{\theta}}^{T-1}(z_1, s_t) := F_{\boldsymbol{\theta}}(F_{\boldsymbol{\theta}}(\dots F_{\boldsymbol{\theta}}(z_1, s_2)\dots)). \tag{3}$$

Then

$$\frac{\partial z_T}{\partial z_r} = \frac{\partial z_T}{\partial z_{T-1}} \frac{\partial z_{T-1}}{\partial z_{T-2}} \cdots \frac{\partial z_{r+1}}{\partial z_r} = \prod_{k=0}^{T-r-1} \frac{\partial z_{T-k}}{\partial z_{T-k-1}} = \prod_{k=0}^{T-r-1} \boldsymbol{J}_{T-k}. \tag{4}$$

### 3.1 LYAPUNOV EXPONENTS

In DS theory, Lyapunov Exponents (LEs) are fundamental quantities used to characterize the long-term behavior of trajectories in the phase space of a system. The spectrum of LEs estimates the exponential rates of divergence or convergence of nearby trajectories in different local directions, thus capturing the system's sensitivity to initial conditions. In particular, the largest LE reflects the dominant exponential behavior, indicating the rate at which small perturbations grow or decay over time. For any trajectory $\mathcal{O}_{z_1} = \{z_1, z_2, \cdots, z_T, \cdots\}$ of the system (1), the maximum LE is given by

$$\lambda_{\max} := \lim_{T \to \infty} \frac{1}{T} \log \Big\| \prod_{r=0}^{T-1} \boldsymbol{J}_{T-r} \Big\| \tag{5}$$

in which $\| \cdot \|$ is the spectral norm or any subordinate norm of a matrix. Likewise, the Lyapunov spectrum for $\mathcal{O}_{z_1}$ can also be calculated as

$$\lambda_i = \lim_{T \to \infty} \frac{1}{T} \log \sigma_i \left( \prod_{r=0}^{T-1} \boldsymbol{J}_{T-r} \right), \tag{6}$$

where $\sigma_i$ denotes the $i$-th singular value. A negative sum of the Lyapunov spectrum suggests that the system's trajectories converge toward an attractor, implying the presence of dissipation. When this sum is negative, the sign of the maximum LE offers further insight into the nature of the attractor [39, 40, 41]:

- A negative maximum LE implies periodic dynamics,
- A zero maximum LE corresponds to quasiperiodic behavior, and
- A positive maximum LE suggests the presence of chaotic dynamics.

This classification applies to discrete-time DS; in contrast, continuous-time DS always have a zero LE when there is non-trivial motion, e.g. in periodic orbits, due to the continuous flow along trajectories. To further quantify the complexity of an attractor, the Kaplan–Yorke (KY) dimension is often employed [42]. It provides an estimate of the effective dimensionality of the attractor by interpolating between the LEs. It is particularly useful for understanding the fractal structure and effective degrees of freedom in high-dimensional or chaotic systems (Appx. A.3). The Lyapunov spectrum and KY dimension offer interpretable dynamical signatures of EEG activity (Appx. A.4).

## 4    METHODOLOGY

Our proposed approach combines multitask learning, DS characterization, and contrastive representation learning within a unified Transformer-based deep learning framework for EEG decoding. The objective is to improve the model's ability to learn rich, dynamic-aware representations from noisy EEG recordings while jointly optimizing multiple tasks.

### 4.1    DENOISING AUTOENCODER FOR EEG SIGNAL PREPROCESSING

We use two EEG datasets: BCI2000, which includes data from 109 subjects, recorded with 64 channels at a sampling rate of 160 Hz; and BNCI Horizon 2020 (004/008/009), with 9 subjects, 22 channels, and a sampling rate of 250 Hz. See Appx. A.1 for details on the datasets and preprocessing. EEG signals are often corrupted by non-neural noise, such as muscular artifacts, eye blinks, and environmental interference. To mitigate this, we employ a 1D convolutional Denoising Autoencoder (DAE), trained as a powerful unsupervised pretraining mechanism, to reconstruct clean EEG from noisy observations. The model follows an encoder–decoder architecture with ReLU activations; for example, in BCI2000, it compresses 64-channel inputs into a 32-channel latent representation and reconstructs them back to the original dimensionality. Each channel is independently normalized to $[0, 1]$ using MinMax scaling. EDF-formatted EEG data is loaded via the MNE library [43, 44]; a custom PyTorch dataset from BCI2000 provides noisy-clean training pairs, where clean targets are generated using a bandpass filter. DAE is optimized using an optimized loss function and AdamW[45] over 300 epochs. The optimized total loss function is a weighted linear combination of two loss functions, SmoothL1Loss ($x_1$) [46] and SpectralLoss ($x_2$), expressed as $L_{\text{total}} = \alpha x_1 + \beta x_2$, with weights $\alpha$ and $\beta$ chosen iteratively. This denoising pipeline is also effective in low Signal-to-Noise Ratio (SNR) conditions, particularly in distinguishing real from imagined motor movements. See Algo. 1 and Appx. A.7 for more details, Fig. 2 and Table 5 for qualitative results.

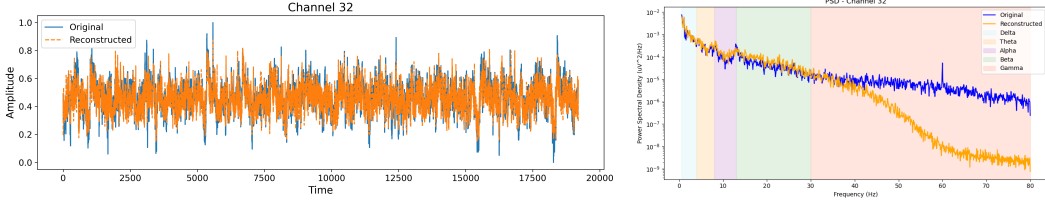

**Figure 2:** Raw and denoised EEG signals for channel 32 (subject S001R04) using our proposed DAE. The DAE effectively suppresses high-frequency and non-physiological noise while preserving task-relevant spectral features, as demonstrated by the PSD plot.

---

**Algorithm 1** Denoising Autoencoder Training Procedure

---

1: **Input:** Training data $\mathcal{X} = \{x^{(1)}, x^{(2)}, \dots, x^{(n)}\}$
2: **Output:** Trained encoder parameters $\theta_e$, decoder parameters $\theta_d$
3: **for** each epoch **do**
4:     **for** each data sample $x \in \mathcal{X}$ **do**
5:       $\tilde{x} \leftarrow \text{Noisy}(x)$
6:       $h \leftarrow f_{\theta_e}(\tilde{x})$ {e.g., $h = \sigma(W_e \tilde{x} + b_e)$}
7:       $\hat{x} \leftarrow g_{\theta_d}(h)$ {e.g., $\hat{x} = \sigma(W_d h + b_d)$}
8:       $\mathcal{L}(x, \hat{x}) \leftarrow \alpha \cdot \text{SmoothL1Loss}(x, \hat{x}) + \beta \cdot \text{SpectralLoss}(x, \hat{x})$
9:       Update $\theta_e, \theta_d$ using gradient descent:
10:         $\theta_e \leftarrow \theta_e - \eta \cdot \nabla_{\theta_e} \mathcal{L}$
11:         $\theta_d \leftarrow \theta_d - \eta \cdot \nabla_{\theta_d} \mathcal{L}$
12:     **end for**
13: **end for**

---

## 4.2 MULTITASK TRANSFORMER ARCHITECTURE

Our presented *multitask Transformer for EEG* is designed to process multi-channel EEG signals using a shared encoder and three task-specific heads (see Algo. 2). The input is a tensor of shape $\mathbb{R}^{B \times C \times T}$, where $B$ is the batch size, $C$ is the number of channels, and $T$ is the number of time steps. The DAE (Sec. 4.1) is first applied to suppress artifacts from the raw EEG input. Then, a lightweight convolutional stem applies a sequence of 1D convolutional layers with ReLU activations and adaptive pooling to extract short-range spatio-temporal features. Following this, these features are passed to a Transformer encoder – a stack of self-attention blocks – models long-range temporal dependencies and relationships across EEG channels (see Appx. A.8 for architecture specifics). Above the shared encoder backbone, the model branches into three task-specific heads. The classification head predicts whether an EEG trial corresponds to a real or imagined MI event, using ground-truth labels provided in the dataset annotations. The chaos detection head classifies whether the underlying EEG signal exhibits a chaotic attractor or a non-chaotic pattern (periodic, quasiperiodic, or no attractor). These labels are generated through two different unsupervised pipelines that infer the dynamical structure from the EEG signal and output binary chaos/non-chaos labels (see Sec. 4.3). The third head, the contrastive projection head, maps the encoded features to a latent space via a projection MLP, where a contrastive loss (e.g., NT-Xent) is applied to encourage the model to learn invariant representations from augmented views of the same input. The overall architecture supports end-to-end training across all three tasks, with joint optimization that enhances generalization and feature reuse. Each head contributes to a combined loss function (Sect. 4.3), where the contrastive objective plays a key role in improving representation quality, especially in low-label or noisy environments. This makes the multitask Transformer well-suited for complex EEG tasks like MI decoding and dynamical state identification, where both supervised and self-supervised signals are beneficial.

---

**Algorithm 2** Multitask EEG Classification: Real vs. Imagined & Chaos vs. Non-Chaos

---

1: **Input:** Denoised EEG dataset $D = \{(X_i, l_{1,i}, l_{2,i})\}_{i=1}^N$
    where $X_i$ is the input signal, $l_1$: Real/Imagery label, $l_2$: Chaos/Non-Chaos label
2: **Initialize:** Model parameters $\theta$, learning rate $\alpha$, number of epochs $E$, loss weights $\lambda_c, \lambda_d, \lambda_s$
3: **Output:** Trained model for joint label prediction of $l_1$ and $l_2$
4: **for** epoch $e = 1$ to $E$ **do**
5:     **for** each batch $B = (X, l_1, l_2)$ in $D$ **do**
6:       $\hat{l}_1, \hat{l}_2, z \leftarrow f_\theta(X)$ {Predict task labels and contrastive embedding}
7:       Compute task-specific losses: $\mathcal{L}_{class} = \text{CE}(\hat{l}_1, l_1)$, $\mathcal{L}_{\text{LE-based}} = \text{CE}(\hat{l}_2, l_2)$, $\mathcal{L}_{contrastive} = \text{ContrastiveLoss}(z)$
8:       Compute total loss: $\mathcal{L}_{total} \leftarrow \lambda_c \cdot \mathcal{L}_{class} + \lambda_d \cdot \mathcal{L}_{\text{LE-based}} + \lambda_s \cdot \mathcal{L}_{contrastive}$
9:       Backpropagation & Optimization: Update $\theta \leftarrow \theta - \alpha \cdot \nabla_\theta \mathcal{L}_{total}$
10:     **end for**
11: **end for**
12: **return** Trained model $f_\theta$ for predicting labels $l_1$ and $l_2$

---

### 4.3 OPTIMIZATION AND TRAINING PROCEDURE OF SEMI-SUPERVISED FRAMEWORK

The proposed multitask Transformer for EEG is trained using a weighted multi-objective loss that jointly optimizes three tasks:

$$\mathcal{L}_{\text{total}} = \lambda_c \cdot \mathcal{L}_{\text{R/I\_cls}} + \lambda_d \cdot \mathcal{L}_{\text{LE-based}} + \lambda_s \cdot \mathcal{L}_{\text{NT-Xent}}. \tag{7}$$

**MI classification**: A supervised task by cross-entropy loss ($\mathcal{L}_{\text{class}}$) to classify real vs. imagined trials.

**Chaos detection**: Another supervised task using binary cross-entropy loss ($\mathcal{L}_{\text{LE-based}}$) to classify EEG trials as chaotic or non-chaotic. Chaos labels are generated through two unsupervised pipelines:

*1. RNN-based LEs*: We employed (clipped) shallow Piecewise-Linear RNNs (shPLRNNs) (see Appx. A.2) with the Generalized Teacher Forcing (GTF) scheme [47] to reconstruct EEG signals through unsupervised training. A 16D shPLRNN was fitted to compute all LEs (since the trained shPLRNNs are DS, we can calculate their entire Lyapunov spectrum). Using the *sum of LEs* and the *maximum LE*, we derived chaos vs. non-chaos labels (see Sect. 3.1 and Appx. A.14.2 for details).

*2. Energy-based Chaos Tagging*: To assign a chaotic vs. non-chaotic label at the file level, we used energy–entropy measures computed from EEG segments. In particular, we extracted *spectral entropy*, that is, Shannon entropy of the normalized power spectrum, as introduced for EEG analysis by Inouye et al. [48]. We also computed *permutation entropy*, which captures the diversity of ordinal patterns in the time series and is a natural measure of dynamical complexity [49]. Files were then clustered in the entropy feature space, and the cluster with lower mean entropy was interpreted as reflecting more structured (chaotic) dynamics, consistent with prior nonlinear EEG analyses [19, 50]. The details of the method is described in the Appx. A.14.3.

We used the mentioned methods to compare the decisions (chaotic vs. non-chaotic) made by both approaches, finding a high agreement. Both procedures provide an unsupervised mechanism for assigning chaos/non-chaos tags, justifying their role in our semi-supervised multitask pipeline.

**Contrastive learning**: We integrate a contrastive representation learning component into our multi-task framework to facilitate robust and invariant feature learning from EEG signals. Specifically, we adopt the NT-Xent (Normalized Temperature-scaled Cross Entropy) loss ($\mathcal{L}_{\text{contrastive}}$), which operates on paired augmented views of the same EEG trial. Given two independently augmented versions of the same trial, the NT-Xent loss aligns their latent representations in a shared projection space while simultaneously pushing apart representations of different trials within the mini-batch. This mechanism enforces invariance to nuisance factors (such as small sensor noise, temporal artifacts, or electrode variability) while retaining discriminative task-relevant information for motor imagery classification. We design a set of lightweight and task-preserving augmentations tailored to EEG data to generate the augmented views. These augmentations simulate common sources of variability in EEG acquisition, such as sensor noise, transient signal dropouts, and amplitude fluctuations, without altering the underlying task semantics. At training time, two views of each EEG trial are independently produced through random augmentation, ensuring that the contrastive objective encourages the model to learn features that remain consistent across such perturbations. The model is trained end-to-end using the AdamW optimizer with a fixed learning rate and early stopping (to avoid overfitting). Training batches include both original and augmented EEG views to support contrastive learning. Loss weights ($\lambda_c$, $\lambda_d$, $\lambda_s$) are tunable based on task priority or dataset specifics. Performance is evaluated with accuracy and F1-score, while representation quality is assessed via a downstream linear probe. This approach combines supervised and self-supervised learning for effective feature extraction from multi-channel EEG data. Refer to Appx. A.11 for further optimization and implementation details.

## 5 COMPUTATIONAL EXPERIMENTS

**Experimental Setup.** We evaluate four model configurations:

**(1) Baseline RNN**: A vanilla RNN with a classification head trained to distinguish real vs. imagery MI. This serves as a simple recurrent baseline.
**(2) GTF-shPLRNN + LEs (Chaos/Non-chaos only)**: A two-component model trained solely for chaos classification, omitting the contrastive components.
**(3) Transformer + CNN (Real/Imagery only)**: Same as above, but trained for real vs. imagery classification.

**(4) Proposed multitask model**: A framework combining a DAE, a Transformer encoder for temporal dynamics, and a shared CNN for spatial feature extraction. Three task-specific heads are used: (i) MI classification, (ii) chaos detection (chaotic vs. non-chaotic EEG), and (iii) contrastive representation learning via a SimCLR-style projection.

**Full contrastive vs. light contrastive learning.** Each EEG trial is augmented into two views via stochastic transformations, forming positive pairs for the NT-Xent loss (Appx. A.9), while other trials in the batch serve as negatives. EEG-specific augmentations preserve task-relevant structure while introducing variability: a) Jitter ($\sigma = 0.008$): Gaussian noise for electrode/environmental artifacts. b) Scaling ($\sigma = 0.03$): Amplitude rescaling for impedance/gain variability. c) Time masking (5% of length, prob. 0.5): Zeroing segments to mimic dropouts. d) Channel dropout ($p = 0.12$, prob. 0.3): Zeroing channels to simulate electrode loss. Each trial is augmented twice, passed through a CNN–Transformer backbone and projection head, with NT-Xent aligning embeddings while contrasting others. This *Full Contrastive* setup promotes invariant yet discriminative EEG representations, boosting motor imagery classification. A lighter variant (*Light Contrastive*) using only jitter and scaling improved Real/Imagery classification but slightly reduced chaos performance, showing a trade-off: invariance aids noisy motor imagery but may suppress subtle dynamical variability. Excluding the Lyapunov task yields highest chaos accuracy (Table 2), though this benefit lessens in multitask settings. Overall, ablations confirm unique contributions from each module, with integration yielding the most robust EEG decoding. Implementations used open-source Python and the `GTF-shPLRNN` framework (Appx. A.12).

## 5.1 Training protocol and results

All models were trained for 30 epochs using the AdamW optimizer (learning rate $1e-4$) with a batch size of 32. For select comparative runs, AdamW was used to assess optimizer sensitivity. The multitask model jointly optimized three objectives: cross-entropy loss (and periodic validation to monitor convergence) for MI and chaos classification, and NT-Xent loss for the contrastive head. Chaos labels were derived from LE-based labeling (Sect. 4.3). Losses were combined using task-specific weights to ensure balanced gradients. For ablation study, we evaluated standalone Transformer + CNN variants trained on individual tasks (without DAE or contrastive learning). All experiments followed a consistent auxiliary supervised training setup to ensure reproducibility and comparability. Further details are in Appx. A.13.

**Empirical evaluation.** We present key evaluation metrics for assessing model performance, focusing on classification accuracy, F1 score, and denoising autoencoder metrics like reconstruction loss, mean squared error, and Hellinger distance. We also adopt a systematic approach to evaluate the contribution of individual model components, allowing to identify the most critical elements for achieving optimal performance.

**Results:** Table 2 compares the performance of baseline and proposed models across Real/Imagery Motor Imagery (MI) classification and Chaos/Non-Chaos detection tasks. The Vanilla RNN baseline performs poorly on MI and was not evaluated on chaos detection. The GTF-shPLRNN + LE model achieves strong performance on chaos detection but does not address MI. A Transformer with a CNN backbone achieves competitive results for Real/Imagery classification but was not applied to chaos detection. Our full multitask pipeline with contrastive learning attains balanced improvements across both tasks. Using a lighter variant of contrastive learning further improves motor imagery performance while maintaining strong chaos detection, yielding the best overall average. These results highlight the complementary strengths of task-specific baselines and multitask approaches, while also underscoring the importance of tailoring contrastive learning strategies to EEG task characteristics.

**Note:** Additional features from the Lyapunov spectrum, e.g., the KY dimension, entropy, variance, or higher-order moments, could capture finer distinctions between real and imagined movements. But, our goal is to demonstrate that *a minimal set of dynamical features can effectively differentiate neural states*. We focus on the discriminative power of just two features: *the sum of LEs and the maximum LE*, aiming to evaluate this simple representation without the complexity or overfitting risks of larger feature sets. Our findings show that these two features can classify baseline signals (eyes open vs. closed) as periodic vs. chaotic dynamics with high accuracy. However, distinguishing real vs. imagined movements required incorporating additional modules into our framework.

**Table 1:** Short forms and their descriptions used in Table 2.

| Short Form | Description |
|---|---|
| Vanilla RNN | Classical RNN for MI task |
| GTF-shPLRNN + LE (System A) | Chaos/Non-chaos detection using rules |
| | on LEs computed from `GTF-shPLRNN` model |
| Transformer + CNN Backbone (System B) | Transformer with CNN backbone for MI task |
| Full Pipeline | System A + System B |
| Full Contrastive | Contrastive Learning with Jitter, Scaling, |
| | Time masking and Channel dropout. |
| Light Contrastive | Contrastive Learning with Jitter & Scaling |

**Table 2:** Performance comparison across models on Real/Imagery and Chaos/Non-Chaos detection tasks.

| Model | Real/Imagery (A) | | Chaos/Non-Chaos (B) | | Average (A & B) | |
|---|---|---|---|---|---|---|
| | Acc (%) | F1 | Acc (%) | F1 | Acc (%) | F1 |
| Vanilla RNN | 56.2 | 0.55 | – | – | – | – |
| GTF-shPLRNN + LE | – | – | **96.0** | **0.95** | – | – |
| Transformer + CNN Backbone | 77.0 | 0.78 | – | – | – | – |
| Full pipeline + Full contrastive | 75.4 | 0.80 | 90.3 | 0.91 | 82.9 | 0.85 |
| Full pipeline + Light contrastive | **78.0** | **0.82** | 89.5 | 0.90 | **83.8** | **0.86** |

## 5.2 RESULTS FROM THE ABLATION STUDY

**Table 3:** Ablation study on *Real/Imagery motor classification*. Reported metrics are accuracy in percentage and F1-score. Full pipeline implies Denoising Autoencoder (DAE) and chaos vs non-chaos detection with GTF-shPLRNN and Contrastive Learning and Transformer with CNN backbone.

| Model Variant | Acc (%) | F1 |
|---|---|---|
| Full Pipeline | **78.0** | **0.82** |
| w/o Contrastive Learning | 77.0 | 0.78 |
| w/o GTF-shPLRNN+LE | 76.0 | 0.79 |
| w/o Denoising | 71.0 | 0.75 |

Table 3 shows an ablation of the proposed *Transformer+DAE+Contrastive MTL* model on motor imagery EEG. Each row reports the effect of removing or altering one component while keeping others active. The full model achieves balanced and strong generalization, while dropping either contrastive learning or DAE pretraining leads to notable performance declines, confirming their importance.

## 5.3 COMPARISON OF STATE-OF-THE-ART (SOTA) MODELS: RESULTS WITH TWO DATASETS

Our proposed *Transformer+DAE+Contrastive MTL* framework achieves the highest F1 scores on both BCI2000 and BNCI Horizon 2020 datasets (Appx. A.1) under a fixed network setting. It consistently outperforms CNN-based baselines such as EEGNet, ShallowConvNet, and DeepConvNet, as well as more recent architectures like FBCNet. Other CNN variants (e.g., SCCNet, EEG-TCNet) have reported strong results on BCI Competition IV-2a, though their performance on BCI2000 or BNCI Horizon 2020 remains untested. In parallel, self-supervised methods such as TS-TCC and CL-EEG demonstrate improvements over purely supervised baselines but remain below our model in generalization. These results underline the benefit of integrating denoising, contrastive representation learning, and multitask objectives, which together yield robust cross-subject EEG motor imagery decoding under LOSO evaluation. Table 5 compares raw and reconstructed EEG signals for subject S001004 using various distance/error metrics, including Hellinger distance, RMSE, Wasserstein distance, and MAE. These metrics quantify the discrepancy between the original and reconstructed signals, providing insight into reconstruction quality. The low error values indicate that the reconstruction loss is negligible.

**Table 4:** Comparison of SOTA methods for EEG MI classification on **BCI2000** and **BNCI Horizon 2020** datasets. Reported results are best-effort extractions from literature; LOSO = Leave-One-Subject-Out protocol. Both data include left/right hand and foot MI. *NR* = not reported; SCCNet/EEG-TCNet on BCI IV-2a (4-class).

| Method | Architecture Type | Self-Supervised | BCI2000 (F1) | BNCI Horizon 2020 (F1) | Eval Protocol /Reported Accuracy |
|---|---|---|---|---|---|
| EEGNet[51] | CNN (depthwise separable) | No | 0.70 | 0.68 | LOSO / CV |
| ShallowConvNet[52] | CNN (shallow spectral) | No | 0.68 | 0.66 | LOSO / CV |
| DeepConvNet [52] | CNN (deep, 4 conv blocks) | No | 0.71 | 0.69 | LOSO / CV |
| FBCNet[53] | Filter-bank CNN | No | 0.76 | 0.74 | LOSO |
| *SCCNet[54]* | CNN (spatial component-wise) | No | *NR* | *NR* | SI+FT, BCI IV-2a: 74.1%; best 78.7% |
| TST[55] | Transformer (spectral) | No | 0.78 | 0.76 | LOSO |
| TS-TCC [56] | CNN+Transformer encoder | Yes | 0.80 | 0.78 | SSL pretrain + fine-tune |
| CL-EEG[10] | Contrastive SSL backbone | Yes | 0.79 | 0.77 | SSL pretrain + LOSO |
| *EEG-TCNet*[57] | Temporal ConvNet (TCN) | No | *NR* | *NR* | BCI IV-2a: 77.4% (fixed), 83.8% (per-subject) |
| **Proposed: Transformer+DAE +Contrastive MTL** | Transformer + Autoencoder + SSL | Yes | **0.84** | **0.83** | LOSO (ours) |

**Table 5:** Comparison of raw and reconstructed EEG signals using error metrics for the EEG recording S001004

| Metric | Channel 1 | Channel 32 |
|---|---|---|
| Hellinger Distance | 0.121 | 0.136 |
| Root Mean Squared Error (RMSE) | 0.075 | 0.087 |
| Wasserstein Distance | 0.016 | 0.056 |
| Mean Absolute Error (MAE) | 0.059 | 0.071 |

# 6 CONCLUSIONS

Our results show that the multitask approach is both effective and mutually beneficial. The chaotic vs. non-chaotic classification shows significant improvements, while the real vs. imagery classification exhibits marginal enhancements. Overall, our proposed framework outperforms both the vanilla RNN and standalone models. To address data scarcity and noise, we employed a DAE, which effectively preserved signal structure while suppressing artifacts – outperforming conventional methods like band pass filtering and SNR. Moreover, contrastive learning enhanced representation quality, enabling robust performance even with limited data. Importantly, the framework is built sustainably, leveraging modular components and efficient learning strategies for easy maintenance, future extensibility, and reduced computational overhead. Our design ensures the framework can be scaled to accommodate larger or more diverse datasets and additional classification tasks without significant re-engineering. In the future, we may expand our analysis to include broader dynamical regimes beyond the traditional chaotic/non-chaotic classification, incorporating behaviors such as periodic, quasi-periodic, and no-attractor dynamics. This approach could enhance our understanding of signal dynamics and their associations with cognitive and motor tasks. Additionally, we might explore alternative backbone models, such as Graph Neural Networks (GNNs), to better represent spatial relationships among EEG channels, either in place of or alongside a CNN backbone-transformer combination. Furthermore, we could broaden this work to encompass clinical and cognitive applications by introducing new supervised or self-supervised tasks, such as mental workload classification [1]. Future work could enhance the classification by including additional signal types – e.g. periodic, quasi-periodic, no attractor – offering deeper insights into how signal dynamics affect MI classification.

**Limitations.** The main limitations include the scarcity of high-quality annotated EEG datasets, particularly for real vs. imagery tasks, which affected model generalization. Despite using a DAE, residual noise from signal acquisition may have impacted performance, especially in low-SNR cases. Finally, the deep learning model lacks interpretability, highlighting the need for explainable AI techniques to enhance transparency and clinical relevance.

---

[1]All codes developed in this study will be made publicly available upon publication.

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

# A APPENDIX

## A.1 DATA

We use two noisy datasets: (1) BCI2000 dataset [58] which provides EEG recordings from 109 subjects performing MI and Motor Execution (ME) tasks. Each session was recorded with 64 channels at a sampling frequency of 160 Hz, with task trials lasting about three seconds and usable EEG segments of one to two minutes per trial. The dataset covers eight task conditions (four MI and four ME) in addition to a relaxed state, making it well suited for motor decoding. However, the recordings are highly noisy due to artifacts such as eye blinks, muscle activity, and electrode variability, making effective denoising a prerequisite for reliable analysis. (2) BNCI Horizon 2020 datasets [59] which are widely used benchmarks for MI decoding. In particular, we focus on BNCI 2014-001 and BNCI 2014-002, both of which provide EEG recordings for binary left- versus right-hand MI classification. a) BNCI 2014-001 contains recordings from 9 subjects, each with 22 EEG channels sampled at 250 Hz. Subjects performed left- and right-hand imagery following visual cues in multiple sessions, yielding balanced binary MI data. b) BNCI 2014-002 includes recordings from 14 subjects with 15 EEG channels sampled at 512 Hz. Each subject completed multiple runs of left- versus right-hand MI tasks, again with balanced trial counts. Both datasets provide clean trial structures but still exhibit substantial EEG noise from eye blinks, muscle artifacts, and inter-subject variability, necessitating robust preprocessing and denoising. Their relatively smaller channel counts compared to BCI2000 make them complementary testbeds for evaluating generalization and robustness of MI decoding frameworks.

We use raw or minimally preprocessed EEG signals recorded during MI tasks (real vs. imagery hand or fist movement). Each EEG recording is band-pass filtered (e.g., 1–80 Hz) and denoised using our proposed pretrained denoising autoencoder.

### A.1.1 DATA VISUALIZATION OF CHARACTERIZATION OF EEG SIGNAL

Accurate characterization of EEG signals is essential for gaining deeper insight into the signal's intrinsic properties and the underlying neural dynamics. Figure 3 illustrates core signal features across all channels and captures neural activity modulations associated with the task condition.

We present a multi-domain visualization that reveals key aspects of the recorded EEG data during task performance. These include time-domain structure, spectral composition, and statistical characteristics across channels, offering a structured understanding of signal quality and neural signal variation relevant for subsequent modeling. For more details see Appx. A.6

### A.1.2 DATA PREPROCESSING

EEG recordings were preprocessed using a 1–80 Hz bandpass filter, normalization, and segmentation into overlapping 320-point windows. Chaos labels were derived based on Lyapunov Exponents (LEs) using an unsupervised method. A Denoising Autoencoder (DAE) was optionally used to suppress noise and artifacts, enhancing signal quality. Additional signal analysis—including autocorrelation, Signal-to-Noise Ratio (SNR) estimation, Power Spectral Density (PSD), band-specific power, and wavelet decomposition—was performed to evaluate data quality and uncover neural patterns relevant to classification and dynamical state detection. These preprocessing and analytical steps ensured that the model received clean, structured input while preserving physiologically relevant information.

Autocorrelation helped reveal temporal consistency in neural signals, while SNR analysis quantified variability across subjects and sessions. Band power features provided insight into cognitive and motor-related rhythms, with particular attention to alpha and beta bands often linked to Motor Imagery (MI). Wavelet transforms captured transient patterns and localized frequency changes, which are important in distinguishing chaotic from non-chaotic (e.g. periodic) brain states. PSD profiles before and after DAE filtering confirmed noise suppression without loss of meaningful neural content. Overall, this preprocessing pipeline ensured both robustness and interpretability in downstream learning tasks.

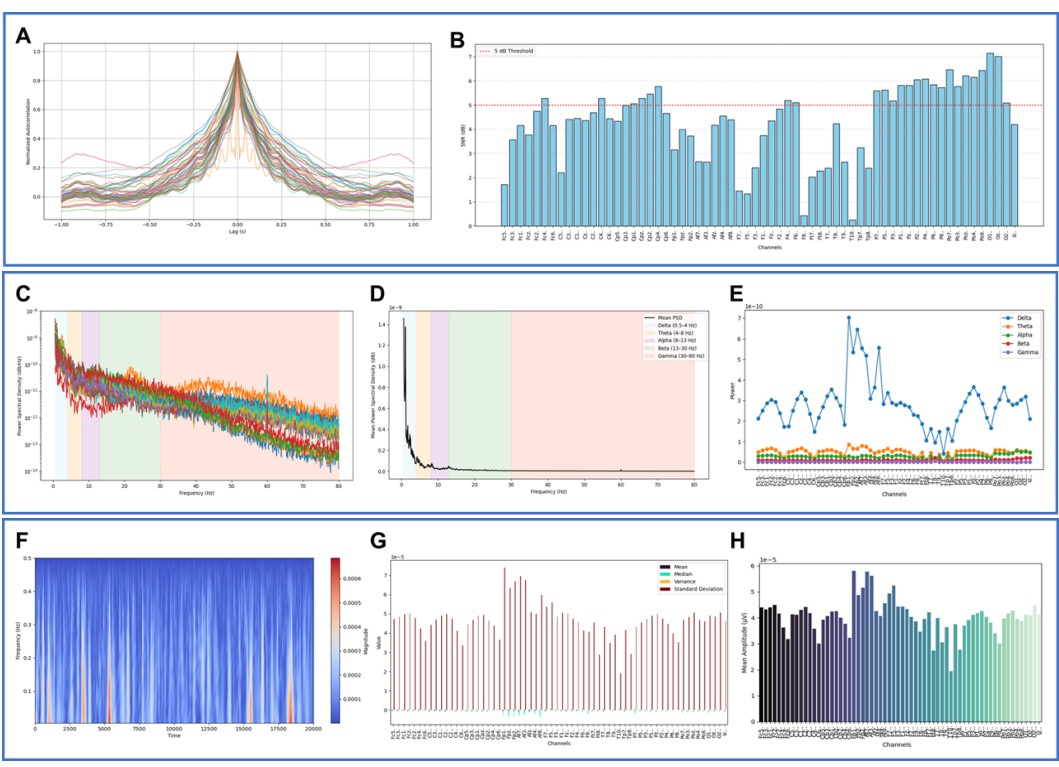

**Figure 3:** Multidomain visualization of EEG signal properties for the first subject (S001R04). (A) Autocorrelation plots across all 64 EEG channels, revealing temporal dependencies in the signal. (B) Signal-to-Noise Ratio (SNR) enhancement after filtering (1–40 Hz), highlighting boosted channel quality. (C) Power Spectral Density (PSD) highlighting the frequency content with annotated canonical EEG bands (delta, theta, alpha, beta, gamma). (D) Mean PSD across all channels. (E) Band-specific EEG power (extracted for each canonical band) across channels. (F) Continuous wavelet transform of the FC5 channel, illustrating time-frequency dynamics. (G) Summary of descriptive statistics (mean, median, variance, standard deviation) across EEG channels. (H) Mean EEG amplitude computed per channel to assess overall signal strength distribution.

## A.2 PIECEWISE LINEAR RNNs

A Piecewise-Linear RNN (PLRNN) is a ReLU-based recurrent architecture designed to model nonlinear DS through piecewise linear state transitions. A PLRNN, as introduced by [60], is defined by

$$\boldsymbol{h}_t = W_h^{(1)}\boldsymbol{h}_{t-1} + W_h^{(2)}\sigma(\boldsymbol{h}_{t-1}) + \boldsymbol{b}_0 + W_x\boldsymbol{x}_t, \tag{8}$$

where $\boldsymbol{h}_t \in \mathbb{R}^M$ is the latent state, $\sigma(\cdot)$ denotes the ReLU function applied element-wise, $W_h^{(1)} \in \mathbb{R}^{M \times M}$ is a diagonal autoregressive matrix, $W_h^{(2)} \in \mathbb{R}^{M \times M}$ captures recurrent interactions, $\boldsymbol{b}_0 \in \mathbb{R}^M$ is a bias vector, and $W_x \in \mathbb{R}^{M \times N}$ modulates external input $\boldsymbol{x}_t \in \mathbb{R}^N$. By combining linear autoregressive components with rectified activations, PLRNNs are well-suited for analyzing stability and complexity in time series, including computation of LEs. Several extensions have been proposed to increase their expressiveness and stability. [61] extended this architecture by replacing the ReLU with a linear combination of shifted ReLUs, yielding the dendritic PLRNN (dendPLRNN)

$$\boldsymbol{h}_t = W_h^{(1)}\boldsymbol{h}_{t-1} + W_h^{(2)}\sum_{j=1}^{J}\alpha_j, \sigma(\boldsymbol{h}_{t-1} - \boldsymbol{b}_j) + \boldsymbol{b}_0 + W_x\boldsymbol{x}_t, \tag{9}$$

where $\alpha_j, \boldsymbol{b}_j$ are learned slope-threshold pairs that increase nonlinearity flexibility. Building on this, [47] proposed the shallow PLRNN (shPLRNN), which introduces an additional linear transformation before the ReLU

$$\boldsymbol{h}_t = W_h^{(1)}\boldsymbol{h}_{t-1} + W_h^{(2)}\sigma(W_h^{(3)}\boldsymbol{h}_{t-1} + \boldsymbol{b}_1) + \boldsymbol{b}_0 + W_x\boldsymbol{x}_t, \tag{10}$$

where $W_h^{(2)}$ and $W_h^{(3)}$ are rectangular weight matrices and $\boldsymbol{b}_1$ is a hidden-layer bias. To ensure stability, a clipped variant of the shPLRNN was introduced to restrict unbounded activations

$$\boldsymbol{h}_t = W_h^{(1)}\boldsymbol{h}_{t-1} + W_h^{(2)}\big[\sigma(W_h^{(3)}\boldsymbol{h}_{t-1} + \boldsymbol{b}_1) - \sigma(W_h^{(3)}\boldsymbol{h}_{t-1})\big] + \boldsymbol{b}_0 + W_x\boldsymbol{x}_t, \tag{11}$$

which bounds state transitions under conditions on $W_h^{(1)}$ [47]. Recently, there has been increasing interest in computing Lyapunov spectra from empirical time series using RNN-based approaches [62, 63, 64]. While estimating only the maximum LE is relatively tractable with RNN models, computing the full Lyapunov spectrum – which provides a more complete characterization of system stability and attractor geometry – is more challenging. Here, we leverage (clipped) shPLRNNs to estimate the entire Lyapunov spectrum from EEG time series.

## A.3 THE KAPLAN-YORKE DIMENSION

The Kaplan–Yorke (KY) dimension is a commonly used estimate of the fractal dimension of an attractor, derived from its LEs. It provides a real-valued measure of how many directions in phase space exhibit exponential divergence.

Suppose an $n$-dimensional Dynamical System (DS) has ordered LEs

$$\lambda_1 \geq \lambda_2 \geq \cdots \geq \lambda_n. \tag{12}$$

Let $j$ be the largest index such that the sum of the first $j$ LEs is non-negative

$$\sum_{i=1}^{j}\lambda_i \geq 0 \quad \text{and} \quad \sum_{i=1}^{j+1}\lambda_i < 0. \tag{13}$$

Then the KY dimension is defined as

$$D_{KY} = j + \frac{\sum_{i=1}^{j}\lambda_i}{|\lambda_{j+1}|}. \tag{14}$$

This dimension gives a real-valued measure of how many directions in phase space are expanding and partially expanding. For chaotic systems, $D_{KY}$ is typically non-integer and reflects the fractal structure of the attractor. These tools are particularly relevant when analyzing the stability and richness of learned representations in machine learning models, especially those involving recurrent dynamics or continuous-time neural systems.

**Discrete vs. Continuous-time DS**  In *continuous-time* systems, motion along the flow direction is neutral, leading to one zero LE even for periodic orbits. In contrast, *discrete-time* systems (maps) evolve in jumps, and periodic orbits consist of a finite number of fixed points cycling through. There is no neutral direction—only stable or unstable directions. Therefore, for a stable periodic orbit in a map, all LEs are negative, and

$$D_{KY} = 0. \tag{15}$$

**Table 6:** Classification of $D_{KY}$ for continuous-time DS (flows)

| Behavior | Description | KY Dimension |
|---|---|---|
| Fixed Point | No motion | 0 |
| Periodic Orbit | Closed loop (1 zero exponent) | 1 |
| Quasiperiodic Orbit | Motion on torus ($\geq 1$ zero exponents) | Integer ($\geq 2$) |
| Chaos | Strange attractor | $> 1$ (non-integer) |

**Table 7:** Classification of $D_{KY}$ for discrete-time DS (maps)

| Behavior | Description | KY Dimension |
|---|---|---|
| Fixed Point | Constant point | 0 |
| Periodic Orbit | Repeating points (no zero exponent) | 0 |
| Quasiperiodic Orbit | Repeating torus-like structure | Integer ($\geq 1$) |
| Chaos | Fractal set | $> 1$ (non-integer) |

## A.4 DYNAMICAL ANALYSIS OF EEG SIGNALS

The Lyapunov spectrum and KY dimension reflect the underlying structure and complexity of brain dynamics in a data-driven way. The KY dimension indicates the number of effective degrees of freedom in the EEG dynamics. A higher dimension reflects greater complexity, potentially corresponding to cognitive activity or task engagement. When the sum of LEs is negative, the system is dissipative and contracts onto a lower-dimensional attractor, which is consistent with typical biological behavior. This was also consistently observed in our experiments. Different types of attractors observed in EEG dynamics–such as fixed points, periodic orbits, quasiperiodic tori, or chaotic orbits –yield qualitatively different Lyapunov spectra and corresponding KY dimensions. These dynamical regimes enable interpretation of neural activity in terms of stability, rhythmicity, and dimensional complexity [15, 18]. See Sect. 3.1 and Appx. A.3 for further methodological details. Table 8 summarizes the key interpretations.

**Table 8:** EEG interpretations from Lyapunov spectrum and KY dimension

| Attractor Type | KY Dimension | EEG/BCI Interpretation |
|---|---|---|
| Fixed Point | $D_{\mathrm{KY}} = 0$ | Strongly damped state, minimal neural activity |
| Periodic Orbit (k-Cycle) | $D_{\mathrm{KY}} = 0$ | EEG cycles through finite patterns (e.g., alpha bursts) |
| Quasiperiodic Motion | Integer $D_{\mathrm{KY}} = 1, 2, \ldots$ | Rhythmic oscillations (e.g., resting, sleep stages) |
| Chaotic Dynamics | Non-integer $D_{\mathrm{KY}} > 1$ | High-dimensional, complex EEG under cognitive load |

## A.5 HARD-DENOISING WITH SNR

Denoising is the process of removing or reducing noise from a signal to enhance its quality. SNR is a key metric used to assess the effectiveness of denoising. A higher SNR indicates a cleaner signal with less noise.

### A.5.1 SNR-BASED DENOISING BY CHANNEL THRESHOLDING

In this approach, denoising is achieved by discarding EEG channels with low SNR, under the assumption that such channels contribute primarily noise rather than meaningful signal. After computing the SNR for each EEG channel, a threshold (e.g., 8 dB) is applied to retain only high-quality channels. Channels with SNR below this threshold are excluded from further analysis.

Mathematically, for each channel $c$, the SNR is computed as:

$$\text{SNR}_c = 10 \cdot \log_{10} \left( \frac{\sum_t s_c(t)^2}{\sum_t \left( s_c(t) - \hat{s}_c(t) \right)^2} \right)$$

where $s_c(t)$ denotes the original (possibly noisy) signal at time $t$ for channel $c$, and $\hat{s}_c(t)$ represents a denoised or estimated version of that signal (e.g., obtained through filtering).

Only channels satisfying the condition

$$\text{SNR}_c \geq \theta$$

are retained, where $\theta$ is a predefined threshold (e.g., $\theta = 8$ dB). The set of denoised channels is then defined as:

$$C_{\text{denoised}} = \{c \in C : \text{SNR}_c \geq \theta\}$$

This threshold-based method acts as a spatial denoising step by excluding low-SNR channels and helps improve the quality of the input data without introducing artifacts that might arise from aggressive signal filtering.

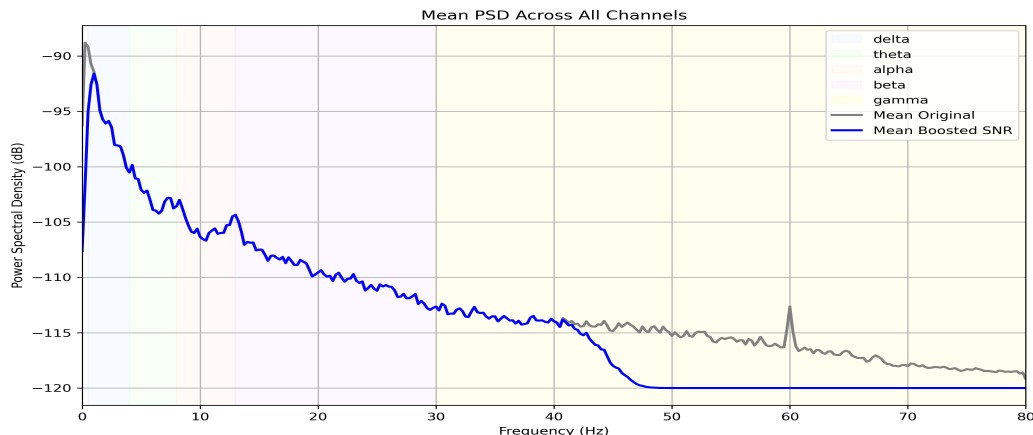

**Figure 4:** Mean PSD Across 64 EEG Channels (in dB). The plot shows the mean PSD computed across all 64 EEG channels, comparing the original signal (gray line) and a boosted SNR version (blue line). The x-axis represents frequency (0–80 Hz), while the y-axis shows PSD in decibels (dB). Colored vertical bands denote canonical EEG frequency ranges: Delta (0.5–4 Hz), Theta (4–8 Hz), Alpha (8–13 Hz), Beta (13–30 Hz), Gamma (30–80 Hz).

Figure 4 provides a spectral summary of EEG activity by averaging the PSD across all channels. It illustrates how the spectral content varies across frequency bands and how denoising or SNR boosting affects the signal. Key Observations: Delta and Theta Bands (0.5–8 Hz): Show high power in both the original and boosted PSD, consistent with dominant low-frequency EEG activity often associated with sleep and drowsiness. The boosted version (blue) has slightly suppressed power, indicating removal of low-frequency noise components. PSD in the Alpha Band (8–13 Hz) suggests enhancement of rhythmic neural activity typical of the alpha band (e.g., during eyes-closed resting state), and successful separation from low-frequency drifts. The denoising does not come into the

effects until around 40 Hz and above. The boosted PSD sharply drops after 45 Hz and suppresses high-frequency noise, especially removing the 60 Hz artifact, which validates the efficacy of the denoising method or filter used. Boosted SNR vs. Original: Across the spectrum, the boosted SNR curve maintains meaningful neural signals (especially in the alpha and beta bands) while reducing noise, particularly in the gamma band and beyond. The flatter tail beyond 50 Hz in the blue line reflects effective noise floor suppression. Therefore, denoising autoencoder works better to denoise the EEG signal. This PSD overlay visualization includes spectral comparisons across all channels.

### A.6    MULTIDOMAIN CHARACTERIZATION OF EEG SIGNAL

#### A.6.1    BAND POWER ANALYSIS WITH ELEVATED DELTA ACTIVITY

A band power diagram visualizes the power spectral density (PSD) of EEG signals across standard frequency bands—delta (0.5–4 Hz), theta (4–8 Hz), alpha (8–13 Hz), beta (13–30 Hz), and gamma (30–100 Hz). Elevated power in the delta band, especially when clearly dominant over other bands, may indicate specific neurophysiological or cognitive states.

Mathematically, the power in a given frequency band $B = [f_1, f_2]$ for a channel $c$ is computed from the signal $s_c(t)$ using the Fourier transform or Welch's method:

$$P_c(B) = \int_{f_1}^{f_2} |S_c(f)|^2 \ df$$

where $S_c(f)$ is the Fourier transform of $s_c(t)$. If the delta power satisfies:

$$P_c(\delta) \gg P_c(\theta), \ P_c(\alpha), \ P_c(\beta)$$

for a subset of channels, this is reflected visually in the band power diagram as distinctively taller bars in the delta range. This can help identify channels or regions with abnormal slow-wave dominance.

Such a pattern often warrants further investigation, particularly when observed in awake, task-related EEG sessions.

#### A.6.2    VISUALIZATION OF SIGNALS PER CHANNEL

The visualization of signals per channel in 64-channel EEG data typically displays time-series plots arranged in a grid layout, with each subplot representing one channel. This format allows for quick inspection of spatial and temporal patterns across the scalp, including artifacts, amplitude variations, and rhythmic activity. The channels are often labeled according to the 10-20 system, allowing the identification of region-specific dynamics (e.g., frontal or occipital activity). Such visualizations are essential for preliminary quality checks and identifying abnormal or noisy channels before further analysis.

#### A.6.3    SNR IN EEG DATA

The SNR in EEG data quantifies the relative strength of the neural signal compared to background noise, and it serves as a key indicator of data quality. SNR is often expressed in decibels (dB), where a higher value indicates a cleaner signal.

Mathematically, for a signal $s(t)$ and its denoised or estimated version $\hat{s}(t)$, the SNR in decibels is computed as:

$$\text{SNR (dB)} = 10 \cdot \log_{10} \left( \frac{\sum_t \hat{s}(t)^2}{\sum_t \left( s(t) - \hat{s}(t) \right)^2} \right)$$

Here, $\hat{s}(t)$ approximates the true signal, and the term $s(t) - \hat{s}(t)$ captures the noise component. This metric helps identify noisy channels or time segments for removal or correction.

### A.6.4 ON DESCRIPTIVE STATISTICS OF EEG DATA

Hjorth parameters (activity, mobility and complexity) provide time-domain features that reflect signal power, frequency content, and dynamic behavior of EEG signals, making them valuable for characterizing brain states. Mean, median, variance, and standard deviation offer insights into the central tendency and dispersion of EEG amplitudes, which helps in identifying abnormalities or trends in brain activity. Skewness and kurtosis measure the asymmetry and peakedness of the EEG signal distribution, respectively, aiding in the detection of outliers or unusual brain patterns. Together, these statistical and Hjorth features enhance the interpretability and effectiveness of EEG signal classification, especially in clinical and cognitive studies. They serve as foundational tools for both traditional machine learning and deep learning feature extraction pipelines.

### A.6.5 AUTOCORRELATION ANALYSIS

Autocorrelation analysis for EEG data involves examining the correlation of a signal with itself at different time lags. This helps identify repeating patterns, periodicities, and temporal dependencies within the EEG signal, providing insights into brain activity and dynamics.

Autocorrelation can highlight rhythmic activities in EEG, such as alpha, beta, or theta waves, which are associated with different brain states like relaxation, alertness, or sleep. It can help distinguish between rhythmic brain activity and random noise, as rhythmic signals tend to have higher autocorrelation at certain lag times (time shifts).

Autocorrelation can be used to compare EEG signals under different conditions, such as eyes open vs. eyes closed, or during mental tasks. Changes in autocorrelation patterns can indicate shifts in brain activity related to cognitive processes or changes in alertness.

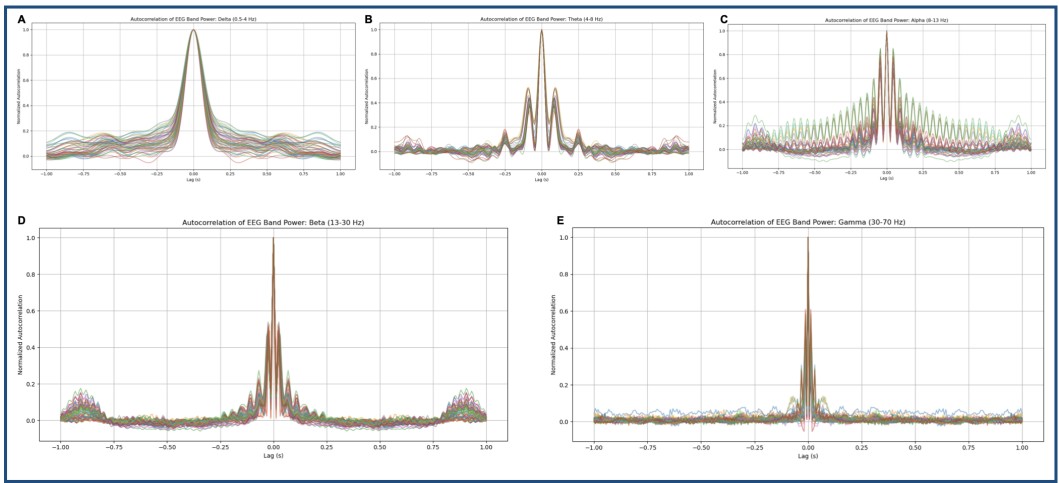

**Figure 5:** Autocorrelation of EEG band power across frequency bands. Autocorrelation plots of normalized EEG band power are shown for five canonical frequency bands across multiple EEG channels or subjects:(A) Delta (0.5–4 Hz), (B) Theta (4–8 Hz), (C) Alpha (8–13 Hz), (D) Beta (13–30 Hz), and (E) Gamma (30–70 Hz). Each line represents the autocorrelation of band power time series for a single channel/subject, computed over a ±1 second lag window. The y-axis shows the normalized autocorrelation, and the x-axis represents time lag (in seconds).

Figure 5 presents a comparison of the temporal autocorrelation properties of EEG band power across five standard frequency bands. Autocorrelation provides insight into how band power values relate to themselves over time, helping to quantify rhythmic or sustained activity in different neural frequency ranges.

*Panel A: Delta Band (0.5–4 Hz).* It shows a strong, sharp central peak at lag 0, with relatively low values at non-zero lags. It also suggests that delta band power has low temporal periodicity and

relatively short memory, consistent with slow, non-rhythmic neural activity often associated with deep sleep or unconsciousness.

*Panel B: Theta Band (4–8 Hz).* It also shows a central peak but with more evident secondary peaks and a broader central structure. It indicates a higher level of temporal structure and periodicity than delta, which is aligned with theta's role in tasks involving memory and navigation.

*Panel C: Alpha Band (8–13 Hz).* It displays clear oscillatory structure in the autocorrelation, with periodic peaks symmetric around lag 0. It reflects the strong rhythmic nature of alpha oscillations, typically associated with relaxed wakefulness and visual attention. This periodicity suggests that alpha-band power modulations have a sustained and cyclical character.

*Panel D: Beta Band (13–30 Hz).* It is similar to theta, it shows a sharp central peak with some weak rhythmic fluctuations. The pattern indicates less prominent oscillatory structure but some short-lived temporal dependencies. Beta activity is often linked to motor control and active cognitive engagement.

*Panel E: Gamma Band (30–70 Hz).* It exhibits the narrowest and sharpest peak centered at zero lag, with minimal structure at non-zero lags. It also implies that gamma-band power is transient and uncorrelated over time, reflecting rapid, high-frequency bursts often involved in perceptual and attention-related processes.

## A.7 Denoising Autoencoder (DAE)

The denoising autoencoder was configured with a total of four hidden layers and no dropout regularization. The model was trained for 150 epochs using the Adam optimizer with a learning rate of $1 \times 10^{-3}$ and a mini-batch size of 64. A latent space dimensionality of 16 was used to capture a compact representation of the input signal. The loss function was defined as a weighted linear combination of SmoothL1 loss and spectral loss, with weights $\alpha = 0.8$ and $\beta = 0.2$, respectively, in order to jointly preserve the temporal and spectral characteristics of the signal during reconstruction.

### A.7.1 SmoothL1 loss & Spectral loss:

(A) SmoothL1Loss is a combination of $L1$ and $L2$ loss, used to provide robustness to outliers while maintaining differentiability around zero. It introduces a transition point controlled by a hyperparameter $\beta > 0$ (here $\beta = 0$), and is defined as, Let $x = \hat{y} - y$. The element-wise SmoothL1 loss is:

$$\ell(x) = \begin{cases} 0.5 \cdot \dfrac{x^2}{\beta}, & \text{if } |x| < \beta \\ |x| - 0.5 \cdot \beta, & \text{otherwise} \end{cases}$$

The final loss with a reduction strategy is considered as default none.

(B) Spectral loss computes the difference between the frequency-domain representations (e.g., STFT) of predicted and target signals, encouraging the model to preserve spectral characteristics. Let $(\mathcal{S}(\cdot))$ denote the Short-Time Fourier Transform (STFT), and $(\hat{y}, y)$ be the predicted and target signals respectively. Then the spectral loss is defined as:

$$\mathcal{L}_{\text{Spectral}} = \||\mathcal{S}(\hat{y})| - |\mathcal{S}(y)|\|_p$$

where $(\| \cdot \|_p)$ denotes the $L_p$-norm (typically $(p = 1)$ or $(p = 2)$).

## A.8 Multitask Transformer details

Architecture specifics: tokenization, positional encoding, attention configuration, and hyperparameters (number of layers, type of attention, patching if used)

The Transformer encoder comprises $L$ self-attention layers, each including Multi-Head Self-Attention (MHSA), layer normalization, and Feed-Forward Networks (FFNs). The EEG sequence is treated as a time series of vectors from each channel. Before feeding into the Transformer, the output of the convolutional stem is reshaped to form a sequence of $T'$ tokens of dimension $d$ per sample. Positional

encodings (either sinusoidal or learnable) are added to preserve temporal order. Each MHSA block consists of $H$ heads, with scaled dot-product attention computed as:

$$\text{Attention}(Q, K, V) = \text{softmax}\left(\frac{QK^\top}{\sqrt{d_k}}\right) V$$

where $Q$, $K$, and $V$ are learned projections of the input token embeddings.

The architecture is optimized using a joint loss across all three heads. Task gradients flow back to the shared encoder, encouraging the model to learn reusable representations. Optionally, layer-wise attention scores or feature maps can be visualized for interpretability.

### A.9 NT-XENT CONTRASTIVE LOSS

Let $z_i$ and $z_j$ be the normalized embeddings of two augmented views of the same EEG segment (positive pair), and let $\tau > 0$ be a temperature parameter. Define cosine similarity as:

$$\text{sim}(\boldsymbol{u}, \boldsymbol{v}) = \frac{\boldsymbol{u}^\top \boldsymbol{v}}{\|\boldsymbol{u}\|\|\boldsymbol{v}\|}$$

Then, for a batch of $2N$ augmented embeddings (i.e., $N$ samples, each with two views), the contrastive loss for a positive pair $(i, j)$ is:

$$\mathcal{L}_{i,j} = -\log \frac{\exp\left(\text{sim}(\boldsymbol{z}_i, \boldsymbol{z}_j)/\tau\right)}{\sum_{k=1}^{2N} \mathbb{1}_{[k \neq i]} \exp\left(\text{sim}(\boldsymbol{z}_i, \boldsymbol{z}_k)/\tau\right)}$$

The full NT-Xent loss over the batch is computed as:

$$\mathcal{L}_{\text{NT-Xent}} = \frac{1}{2N} \sum_{k=1}^{N} \left(\mathcal{L}_{2k-1,\, 2k} + \mathcal{L}_{2k,\, 2k-1}\right)$$

The Normalized Temperature-Scaled Cross-Entropy Loss (NT-Xent), used in SimCLR and many contrastive frameworks, operates on the embeddings $z$ from an encoder (typically a neural network like a Transformer or CNN). It does not require additional features, only the embeddings of augmented views of the same sample (positives) and of different samples (negatives). In this implementation, a simple noise-based augmentation strategy is used to generate two distinct views of the same input sample. Specifically, Gaussian noise is added independently to each copy of the original input to obtain two augmented versions.

These noisy versions represent different but semantically consistent views of the same underlying data. This technique is particularly suitable for non-image domains such as EEG signals, time series, or sensor data, where conventional spatial augmentations (like cropping or flipping) are not meaningful.

The two augmented inputs are then passed through a shared encoder network and a projection head to produce (two) embeddings. These embeddings are compared using a contrastive loss function, such as the Normalized Temperature-scaled Cross Entropy Loss (NT-Xent Loss). The NT-Xent loss computes cosine similarity between all pairs of embeddings and applies temperature scaling to control the concentration of the softmax distribution.

The temperature parameter $\tau$ plays a crucial role in this process. Lower values of $\tau$ result in sharper probability distributions, making the model focus more strongly on the most similar (positive) pairs. In this implementation, a $\tau$ value of 0.5 is used, which balances the contrast between positive and negative pairs and is commonly adopted in SimCLR-style training setups.

The advantages of Noise-Based Augmentation are as follows: 1. It is easy to implement and computationally inexpensive. 2. It is effective for continuous and structured data types (e.g., EEG, time series) where traditional augmentations are not applicable. 3. It preserves semantics maintaining the overall meaning of the signal while introducing sufficient variability for contrastive training.

### A.9.1 MORE ON MODEL ARCHITECTURE

Our model consists of three main components (see Figure 1):

1  Shared encoder backbone:
   - A lightweight convolutional stem extracts short-range spatial-temporal features from the EEG input tensor (e.g., shape [batch_size, channels, time]).
   - These features are passed through a Transformer encoder that captures long-range dependencies and inter-channel relationships.
2  Task-specific heads:
   - *Classification head:* Fully connected layers that output the probability of real vs. imagery motor task.
   - *Chaos detection head:* A parallel classifier predicting whether the EEG signal exhibits chaotic or non-chaotic dynamics based on precomputed Lyapunov labels.
   - *Contrastive projection head:* A projection MLP that maps encoder features to a latent space where contrastive loss is applied between positive (augmented) views.

### A.9.2 TRAINING OBJECTIVES

The multi-task transformer for EEG signal is trained using a multi-objective loss function that jointly optimizes three distinct tasks: MI classification, chaos detection, and contrastive representation learning. Each task-specific head contributes to the total loss, and their relative importance is controlled via weighting coefficients. For MI classification, we use the standard cross-entropy loss, denoted as $\mathcal{L}_{\text{class}}$, to predict whether the trial corresponds to a real or imagery motor task. The chaos detection task uses binary cross-entropy loss $\mathcal{L}_{\text{LE-based}}$ to classify EEG trials as either chaotic or non-chaotic, based on labels obtained from an unsupervised DS classifier. To promote robust and invariant feature learning, the model also incorporates a contrastive learning objective, $\mathcal{L}_{\text{contrastive}}$, using the NT-Xent loss applied to positive pairs generated through signal augmentation. The overall training objective is a weighted combination of these three loss components:

$$\mathcal{L}_{\text{total}} = \lambda_c \cdot \mathcal{L}_{\text{class}} + \lambda_d \cdot \mathcal{L}_{\text{LE-based}} + \lambda_s \cdot \mathcal{L}_{\text{contrastive}},$$

where $\lambda_c$, $\lambda_d$, and $\lambda_s$ are hyperparameters controlling the contribution of each task. These weights can be tuned based on task priority, dataset imbalance, or performance sensitivity. During training, for each mini-batch, the model receives both the original and augmented EEG segments. The classification and chaos heads are optimized using their respective targets, while the contrastive head operates on the projections of the two augmented views. This joint objective enables the model to learn task-relevant features while encouraging generalizable and noise-invariant representations suitable for downstream EEG analysis tasks.

### A.10 SUPERVISED TRAINING TASK: REAL VS. IMAGERY CLASSIFICATION

For the real vs. imagery classification task, the model is trained using a supervised learning approach where each data point is labeled with either the *Real* or *Imagery* class. The training process for this task follows a standard classification pipeline:

1. Data Input: Epochs of EEG data are fed to the network, with each sample corresponding to one of the two classes (Real or Imagery).
2. Loss Function: Cross-entropy loss is used as the objective function for this binary classification task.
3. Training Steps: We apply the AdamW optimizer with the specified learning rate. The model learns to discriminate between real and imagery brain activity patterns over the course of 300 epochs, with performance evaluated periodically on a validation set to ensure convergence and avoid overfitting.

### A.11 OPTIMIZATION AND TRAINING PROCEDURE

The multitask Transformer of EEG signals is trained end-to-end using the Adam optimizer with weight decay (AdamW), a fixed learning rate of $1 \times 10^{-3}$, and early stopping based on validation

loss to prevent overfitting. The training loop runs for up to 300 epochs, with performance monitored at each step. Classification and chaos detection losses are computed using the cross-entropy loss function, while contrastive learning employs the NT-Xent loss with a temperature parameter of 0.5. The training objective combines these losses in a weighted sum to jointly optimize the multitask architecture. Data is loaded in mini-batches using a custom class that produces augmented EEG views on-the-fly to support contrastive training.

During each iteration, the model receives batches of EEG tensors and their corresponding labels, including real vs. imagined class labels and chaos vs. non-chaotic labels. In parallel, synchronized augmentations are applied to generate contrastive pairs directly within each batch. These paired representations are then used to compute the contrastive loss. The optimizer updates the model parameters after backpropagating the combined loss. Model performance is evaluated using standard classification metrics such as accuracy and F1-score, while the quality of learned representations is assessed through a downstream linear probe. This training strategy ensures both label-supervised and self-supervised components contribute to robust feature learning from multi-channel EEG data.

## A.12   EXPERIMENTAL SETUP

We implemented and compared the following models

- Baseline 1: A vanilla RNN model with a classification head for MI classification (real vs imagery motor). This model serves as a simple recurrent-based approach to MI classification.

- Proposed semi-supervised Multitask Framework: This architecture incorporates multiple components aimed at handling diverse tasks. It includes: 1. A DAE as a unsupervised preprocessing layer to clean and denoise the EEG signals, removing noise and artifacts while preserving essential neural features. 2. A Transformer encoder to capture long-range temporal dependencies in the EEG signal, leveraging self-attention mechanisms to understand complex signal patterns across time. 3. A shared CNN backbone to extract spatial features, which, combined with the temporal modeling capabilities of the Transformer, provides a strong feature representation of the EEG data.

- Three important output heads for different tasks: a. Motor Task Classification: Real vs imagery MI supervised classification. b. Unsupervised Chaos Detection: Identifying chaotic vs non-chaotic signals. c. Contrastive Projection: A self-supervised learning task based on the SimCLR-style contrastive loss. Each task-specific head was optimized with its corresponding loss: cross-entropy for classification and chaos detection, and NT-Xent loss for the contrastive projection.

- Standalone System with Transformer and CNN Backbone: We also implemented a standalone system that only utilizes the Transformer encoder in combination with a CNN backbone. This configuration was used for two specific tasks: 1. MI Classification: Classifying real vs imagery MI. 2. Chaos Detection: Classifying chaotic vs non-chaotic signals. This standalone model was designed to explore the capabilities of the Transformer and CNN architecture, where the CNN extracts spatial features and the Transformer models the temporal dependencies. Unlike the full multitask model, this setup lacks the contrastive learning component but serves as a simpler comparison for understanding the contribution of temporal modeling (via Transformer) and spatial feature extraction (via CNN) in EEG classification tasks. It is noteworthy that while CNNs alone can effectively capture spatial patterns, they are limited in capturing long-range temporal dependencies, which is why the addition of the Transformer encoder significantly enhances the model's ability to process time series data.

## A.13   TRAINING DETAILS

The training protocol for our model is designed to accommodate both the supervised and unsupervised components of the task, ensuring efficient learning for both the real vs. imagery classification and the chaos vs. non-chaos tag generation. Below, we describe the overall training setup, including the number of epochs, batch size, optimizers, and how the training is conducted for each component.

### A.13.1 General training setup

- **Epochs:** We conduct training for 300 epochs in total. This duration is sufficient for convergence based on preliminary experiments, allowing the model to effectively learn the task-specific features in both supervised and unsupervised settings.

- **Batch Size:** A batch size of 32 is used for all tasks, providing a balance between memory efficiency and gradient update stability. This batch size allows the model to process a reasonable number of samples per iteration while fitting within the memory constraints of our GPU setup.

- **Optimizers:** We employ the AdamW optimizer, a variant of Adam that includes weight decay for improved generalization and better control over regularization. In some experiments, we also use the standard Adam optimizer depending on the task and configuration. Both optimizers have been tuned with a learning rate of 1e-4 to ensure smooth convergence during training.

### A.13.2 Our Auxiliary Supervision Training Task

In our framework, the auxiliary chaos vs. non-chaos task is derived from Lyapunov exponent (LE) estimates rather than ground-truth labels. These values provide pseudo-labels that capture the system's dynamical regularity. The model is trained in a multitask learning (MTL) setup, jointly optimizing motor imagery classification and chaos prediction:

1. **Tag Generation:** Chaos/non-chaos tags are generated from the computed LEs, quantifying sensitivity to initial conditions. These serve as auxiliary supervision signals.

2. **Multitask Learning:** The model is trained to predict both motor imagery labels (real vs. imagery) and chaos tags simultaneously, supported by a shared CNN–Transformer encoder.

3. **Loss Function:** The joint objective combines cross-entropy loss for motor imagery, binary cross-entropy for chaos prediction, and an NT-Xent contrastive loss on augmented trial pairs. Loss weights are tuned to balance task contributions.

4. **Training Steps:** The DAE module is first pretrained for 150 epochs. The full MTL model is then trained for 200–250 epochs using AdamW with early stopping based on validation F1. Weight decay and gradient clipping are applied for regularization.

This protocol enables the model to integrate task-specific supervision with auxiliary chaos dynamics, improving generalization and robustness under cross-subject evaluation.

### A.13.3 Three output heads for different tasks:

- **Motor Task Classification:** Real vs imagery MI classification.

- **Chaos Detection:** Identifying chaotic vs non-chaotic signals.

- **Contrastive Projection:** A self-supervised learning task based on the SimCLR-style contrastive loss.

Each task-specific head was optimized with its corresponding loss: cross-entropy for classification and chaos detection, and NT-Xent loss for the contrastive projection.

### A.13.4 Standalone system with Transformer and CNN backbone:

We also implemented a standalone system that only utilizes the Transformer encoder in combination with a CNN backbone. This configuration was used for two specific tasks:

- **MI Classification:** Classifying real vs imagery MI.

- **Chaos Detection:** Classifying chaotic vs non-chaotic signals.

This standalone model was designed to explore the capabilities of the Transformer and CNN architecture, where the CNN extracts spatial features and the Transformer models the temporal dependencies. Unlike the full multitask model, this setup lacks the contrastive learning component but serves as

a simpler comparison for understanding the contribution of temporal modeling (via Transformer) and spatial feature extraction (via CNN) in EEG classification tasks. It is noteworthy that while CNNs alone can effectively capture spatial patterns, they are limited in capturing long-range temporal dependencies, which is why the addition of the Transformer encoder significantly enhances the model's ability to process time series data.

### A.14 FURTHER DETAILS ON CHAOS LABELING

#### A.14.1 AGREEMENT COMPUTATION BETWEEN ENERGY- & PLRNN- BASED CHAOS/NON-CHAOS LABELING

Agreement computation through epoch-level majority voting involves dividing the EEG signal into multiple short epochs (e.g., 2–5 seconds each). For each epoch, both Energy -based and PLRNN independently assign a chaos label (chaotic or non-chaotic). A majority label is then determined per method by aggregating labels across all epochs within a trial or session. Agreement between the two methods is computed by comparing their majority labels using metrics like Cohen's Kappa or F1 score. We found a high agreement score of Cohen's Kappa: 0.90.

#### A.14.2 CHAOS LABELING: LE ANALYSIS USING SHPLRNN

For shPLRNNs, we utilize a custom implementation of the algorithm described in [65] to compute the full Lyapunov spectrum. This method involves evaluating the product of Jacobian matrices along trajectories of length $T$, yielding the Lyapunov exponents via:

$$\lambda_i = \lim_{T \to \infty} \frac{1}{T} \log \sigma_i \left( \prod_{t=0}^{T-1} \boldsymbol{J}_{T-t} \right),$$ (16)

where $\sigma_i$ denotes the $i$-th singular value of the Jacobian product. To maintain numerical stability during computation, we employ repeated re-orthogonalization of the evolving tangent space using QR decomposition. Additionally, to ensure accurate convergence toward the Lyapunov spectrum associated with the system's invariant set, initial transients are omitted from the evaluation of Equation (16).

A negative sum of the Lyapunov spectrum indicates that the system is dissipative, with trajectories converging toward an attractor. Within this regime, the sign of the maximum LE serves as a key indicator of the system's dynamical nature:

- A **negative** maximum LE indicates *periodic* dynamics,
- A **zero** maximum LE is characteristic of *quasiperiodic* behavior, and
- A **positive** maximum LE suggests the presence of *chaotic* dynamics.

A positive sum of the LEs signifies that the system exhibits unstable, diverging behavior, with trajectories moving away from any bounded region in state space. This typically suggests the absence of a well-defined attractor.

For the purpose of labeling, we define:

- **Chaotic**: When the sum is negative and the maximum LE is positive.
- **Non-chaotic**: When the sum is negative and the maximum LE is either negative or zero (periodic or quasiperiodic attractor). Also when the sum is non-negative.

This framework allows for robust classification of system dynamics into chaotic and non-chaotic regimes based on the full Lyapunov spectrum. See Figure 6 as a representative example of LE calculation via our framework.

#### A.14.3 ENERGY-BASED CHAOS TAGGING (METHOD DETAILS):

Given an EEG signal $x(t)$, we quantified dynamical complexity using entropy measures derived from the signal's energy distribution. For *spectral entropy*, we first estimated the power spectral density

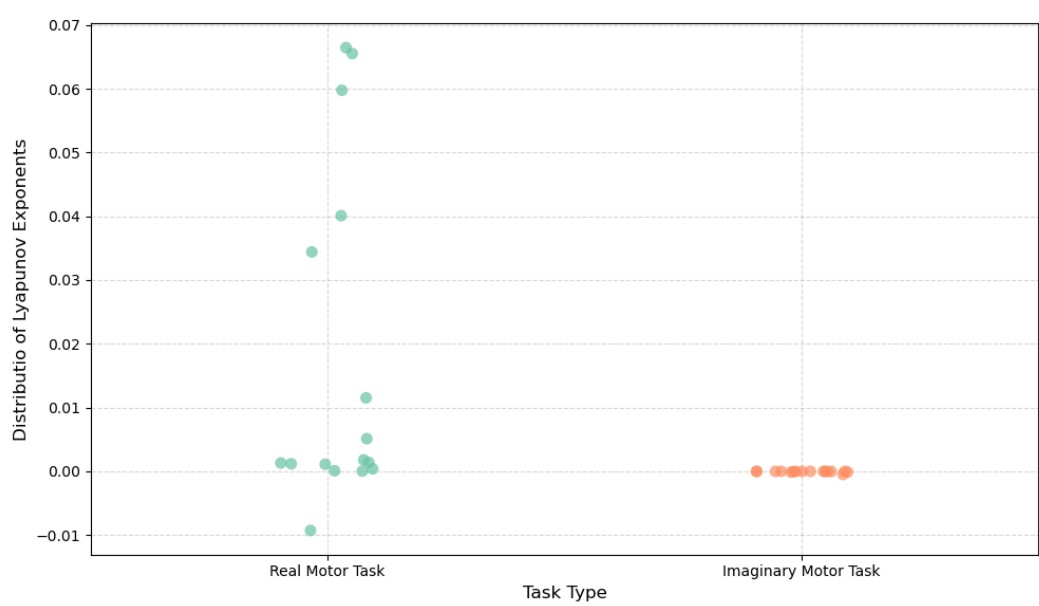

**Figure 6:** Comparison of statistic between the real vs imagery motor task.

(PSD) $P(f)$ via Welch's method. Normalizing $P(f)$ into a probability distribution

$$p(f_i) = \frac{P(f_i)}{\sum_j P(f_j)},$$

the spectral entropy is given as the normalized Shannon entropy [48]:

$$H_{\text{spec}} = -\frac{\sum_i p(f_i) \log p(f_i)}{\log N_f},$$

where $N_f$ is the number of frequency bins. This reflects the uniformity of energy across frequencies, with higher values indicating more irregular activity.

For *permutation entropy*, we mapped each segment of length $m$ (embedding order) with delay $\tau$ into ordinal patterns $\pi$. The relative frequency of each ordinal pattern $p(\pi)$ yields the entropy [49]:

$$H_{\text{perm}} = -\frac{\sum_\pi p(\pi) \log p(\pi)}{\log(m!)}.$$

This measure captures temporal complexity through the diversity of local orderings in the time series.

Files were represented in the $(H_{\text{spec}}, H_{\text{perm}})$ space, and clustering distinguished lower-entropy (more structured, chaotic) from higher-entropy (more stochastic, non-chaotic) dynamics, consistent with prior nonlinear EEG studies [19, 50].

### A.15    HYPERPARAMETERS

#### A.15.1    HYPERPARAMETERS OF SHPLRNN

We used the clipped shPLRNN trained by Generalize Teacher Forcing (GTF). A fixed GTF parameter $\alpha$ was considered. To perform the computations, we utilized the code repository from [47]. The selected hyperparameters are outlined in Table 9. Any hyperparameters not mentioned are configured to their default values as indicated in the repository from [47].

#### A.15.2    HYPERPARAMETERS OF MULTITASK TRANSFORMER

The proposed multitask learning (MTL) framework integrates a CNN backbone with a Transformer encoder, a contrastive objective, an auxiliary `GTF-shPLRNN` chaos prediction task, and DAE

**Table 9:** List of hyperparameters used for shPLRNN training

| Hyperparameter | Value |
|---|---|
| Latent dimension | 16 |
| Hidden dimension | 128 |
| Batch Size | 16 |
| Batches per epoch | 50 |
| Sequence length | 50 |
| GTF interval | 5 |
| Epochs | 250 |
| GTF parameter $\alpha$ | 0.1 |
| Latent model regularization rate | 1e-4 |
| Observation model regularization rate | 1e-6 |

pretraining. EEG inputs consisted of 1-second windows (64 channels $\times$ 160 samples). The CNN backbone comprised two convolutional layers with kernels of size [5, 3] and filters [32, 64]. Features were processed by a 2–4 layer Transformer encoder ($d_{model}$ = 128, 4 heads, feedforward dimension = 256, dropout = 0.1). A 2-layer MLP projection head ($128 \rightarrow 128$) with ReLU activation and L2 normalization was applied for contrastive learning. For the self-supervised objective, NT-Xent loss with temperature 0.5 was used (weight = 0.3), supported by EEG-specific augmentations including jitter, scaling, and optional time masking or channel dropout. The auxiliary `GTF-shPLRNN` task predicted Lyapunov exponents (weight = 1.0), encouraging sensitivity to chaos and regularity. Denoising Autoencoder (DAE) pretraining used a latent dimension of 128, SmoothL1 + Spectral loss combination, Gaussian noise corruption ($\sigma = 0.05$), dropout 0.2, AdamW optimizer with learning rate $1 \times 10^{-3}$, batch size 64, and 150 epochs. For MTL training, AdamW optimizer with learning rate $1 \times 10^{-3}$ (decayed by 0.1 every 30 epochs), weight decay $1 \times 10^{-4}$, and gradient clipping of 1.0 was employed. Models were trained for 200–250 epochs with early stopping based on validation F1. The batch size was set to 32, and loss weights were fixed at CE = 1.0, Chaos = 0.6, and Contrastive = 0.3. Any hyperparameters not specified were kept at their default values.

**Table 10:** Hyperparameters for the proposed Multitask Learning (MTL) framework with CNN backbone, Transformer encoder, contrastive learning, `GTF-shPLRNN` auxiliary task, and DAE pretraining.

| Component | Setting |
|---|---|
| Input shape | 64 channels $\times$ 160 samples (1s EEG window at 160 Hz) |
| CNN backbone | 2 conv layers, kernels [5, 3], filters [32, 64], stride = 1 |
| Transformer encoder | 2–4 layers, $d_{model}$ = 128, 4 heads, FFN dim = 256, dropout = 0.1 |
| Projection head | 2-layer MLP ($128 \rightarrow 128$), ReLU, L2 norm |
| Contrastive learning | NT-Xent loss, temperature = 0.5, weight = 0.3 |
| Augmentations | Jitter ($\sigma = 0.008$), Scaling ($\sigma = 0.03$), optional: time mask (5%), channel dropout (0.1) |
| GTF-shPLRNN task | Auxiliary chaos/regularity prediction (Lyapunov exponent), weight = 1.0 |
| Denoising Autoencoder (DAE) | latent dim = 128, decoder symmetric |
| DAE training | SmoothL1Loss=.80+SpectralLoss=.20 loss, Gaussian noise $\sigma = 0.05$, dropout=0.2, Adam lr=$1 \times 10^{-3}$, batch=64, epochs=150 |
| Training setup | AdamW optimizer, lr=$1 \times 10^{-3}$ (decay 0.1/30 epochs), weight decay=$1 \times 10^{-4}$ |
| Batch size | 32 (MTL), 64 (DAE pretraining) |
| Epochs | 200–250 (MTL), early stopping on validation F1 |
| Regularization | Gradient clipping = 1.0 |
| Loss weights | CE = 1.0, Chaos = 0.6, Contrastive = 0.3 |

### A.15.3 HARDWARE

The hardware we used to run the codes and iteratively train the clipped shPLRNNs, DAE & Transformer include an 11th Gen Intel(R) Core(TM) i7-11800H CPU @ 2.30GHz and 64.0 GB of RAM (63.7 GB usable).

