# OpenReview forum: "Contrastive and Multi-Task Learning on Noisy Brain Signals with Nonlinear Dynamical Signatures"
_ICLR.cc/2026/Conference — ICLR 2026 Conference Withdrawn Submission_

### Official Review · Reviewer_BLEy · 2025-10-20

**Soundness:** 3
**Presentation:** 3
**Contribution:** 2
**Rating:** 2
**Confidence:** 4

**Summary:**

A neural decoding model that combines multitask learning with contrastive learning. As a part of tasks, predicting dynamicity signatures (chaos vs non-chaos) by Lyapunov Exponents is included and show improvements in decoding accuracy, suggesting the effectiveness of the proposed multitask training including the dynamics objective.

**Strengths:**

* Adoption dynamic theory in a multitask training setting for neural decoding which is shown effective.

* Achieved state-of-the-art performance in motor imagery classification task using EEG.

**Weaknesses:**

* While the decoding results are promising, the ideas of multitask training and contrastive learning using NT-Xent are already well-proven and frequently deployed methodologies in the field for years. Adopting the LE-based loss is an interesting and novel idea, but the major source of performance gain is driven by denoising as shown in Table 3.

* Moreover, integrating dynamic state theory is interesting but the main results and discussions are mostly focused on the decoding performance. This can provide some potential novel interpretation of the modeling of EEG. I encourage to including more insightful analysis that can better leverage this idea to demonstrate scientific utility of the proposed method beyond improving performance.

**Questions:**

No specific question.

---

### Official Review · Reviewer_V5Xr · 2025-11-02

**Soundness:** 2
**Presentation:** 1
**Contribution:** 1
**Rating:** 2
**Confidence:** 3

**Summary:**

The paper proposes a two-stage multitask learning framework for EEG signal analysis that integrates denoising, dynamical modeling, and representation learning. The multitask architecture incorporates three objectives: motor imagery classification, chaotic versus non-chaotic regime discrimination, and self-supervised contrastive representation learning. Experimental results demonstrate its superiority over recent state-of-the-art methods.

**Strengths:**

- The paper is clearly organized and easy to follow.

**Weaknesses:**

- The contribution of this paper appears unclear to me. The multitask design is somewhat confusing, and further clarification is needed on the motivation for incorporating motor imagery (MI) classification into the original contrastive self-supervised representation learning framework. Moreover, the determination of whether signal dynamics are chaotic or non-chaotic through Lyapunov exponent estimation requires stronger theoretical justification and empirical evidence to demonstrate its effectiveness in capturing meaningful temporal dynamics and improving decoding performance.

- The schematic illustration presented in Figure 1 lacks sufficient detail to clearly convey the proposed contributions. More explicit formulations are needed to describe how each loss term, such as L_constrastive is computed. The experimental evidence provided is also not sufficiently strong; additional datasets and evaluation metrics should be included to more convincingly demonstrate the superiority of the proposed multitask architecture. Furthermore, a more comprehensive analysis of the chaotic temporal dynamics is warranted.

**Questions:**

- What is the role of the ConvNet illustrated in Figure 1? Is it integrated as part of the Transformer encoder or used as a separate feature extractor?

- How is the NT-Xent loss computed, and what strategy is used to generate augmented views of the EEG signals?

- How does discriminating between chaotic and non-chaotic trials contribute to improving behavioral decoding performance?

- Considering that Lyapunov exponents (LE) can be directly estimated from the original signals, what is the necessity of fitting a specific model to perform this estimation?

---

### Official Review · Reviewer_4QHr · 2025-11-07

**Soundness:** 2
**Presentation:** 2
**Contribution:** 2
**Rating:** 6
**Confidence:** 3

**Summary:**

This paper presents a two-stage framework for EEG decoding that integrates denoising, multitask learning, and contrastive self-supervision. The use of Lyapunov exponent–based chaos detection is a novel approach that bridges nonlinear dynamical systems theory with deep learning, introducing an additional layer of interpretability to the model’s behavior. Experiments on the BCI2000 and BNCI Horizon datasets show steady improvements over strong baselines, and the ablation studies clearly highlight the value of each component in the overall design. The approach is technically solid and conceptually innovative, though the study would benefit from testing on a wider range of datasets and including details on training time and efficiency. Overall, this work effectively combines insights from neuroscience and machine learning to enhance EEG analysis.

**Strengths:**

* The paper demonstrates originality by introducing a combination of denoising, multitask learning, and contrastive self-supervision within a unified EEG decoding framework.

* The incorporation of Lyapunov exponent–based chaos detection is particularly novel, as it bridges nonlinear dynamical systems theory with modern deep learning, an intersection rarely explored in this domain.

* The architectural design is well-motivated.

* The paper is clearly written, with a logical flow that makes complex concepts, such as Lyapunov analysis, accessible to a broader audience.

* The proposed approach advances EEG signal modeling by improving both robustness and interpretability, and it has potential implications for neuroscience-inspired machine learning more broadly.

**Weaknesses:**

* The experimental validation is limited to two datasets (BCI2000 and BNCI Horizon 2020), both focused on motor imagery, which constrains the generalizability of the results.

* Including additional datasets, such as BCI Competition IV-2a, PhysioNet EEG Motor Movement, or emotion and clinical EEG corpora, would better demonstrate cross-domain robustness.

* The paper does not report computational metrics such as training time, parameter count, or inference latency, which are important for assessing scalability and reproducibility.

* The chaos detection task relies on Lyapunov exponent–derived labels rather than empirical ground truth, introducing potential circularity in evaluation. The authors could strengthen this aspect by validating chaos labels against known nonlinear EEG benchmarks (e.g., Bonn or CHB-MIT datasets).

* The paper would benefit from more interpretability analysis, such as attention visualizations or feature importance studies, to better illustrate what the model learns about neural dynamics.

Addressing these points would make the work more comprehensive, reproducible, and broadly convincing.

**Questions:**

1. Could the authors elaborate on how consistent the Lyapunov exponent–based chaos/non-chaos labels are across subjects and datasets? For instance, what is the variability in label assignment when using RNN-derived Lyapunov exponents versus entropy-based clustering? Providing a quantitative agreement measure (e.g., Cohen’s κ or correlation) would clarify the stability of the labels.


2. Do the authors expect the framework to generalize to other EEG paradigms, such as emotion classification or seizure detection?


3. What are the training times and resource requirements for each stage (DAE pretraining and multitask Transformer)? A runtime comparison against baselines, such as EEGNet or DeepConvNet, would help readers assess the practical trade-offs between performance and complexity.


4. How sensitive is the framework to the selection of augmentations and temperature parameters in the NT-Xent loss?

**Details Of Ethics Concerns:**

NA.

---

### Official Review · Reviewer_rqr5 · 2025-11-11

**Soundness:** 1
**Presentation:** 3
**Contribution:** 2
**Rating:** 2
**Confidence:** 3

**Summary:**

The paper proposes a new framework for motor-related EEG decoding that is build on denoising autoencoders, multitask training (motor-related classification. chaotic vs non-chaotic discrimination and contrastive learning). They evaluate their hybrid convolutional-transformer-architecture trained in this way on two EEG motor-related datasets and report improved performance over competing decoding methods.

**Strengths:**

*  Novel idea of using chaotic vs nonchaotic as a pretext task
* ablation study to show effect of individual components
* I found the writing to be fairly understandable

**Weaknesses:**

**Result inconsistencies**

This work proposes a fairly complicated two-stage multi-task architecture for EEG decoding. To justify the complexity of the pipeline, the empirical evaluation must be very rigorously done and show where the pipeline improves over simpler decoding pipelines. Unfortunately, there seem to be multiple fundamental errors that question the results, especially the reported results of other methods in the manuscript:

In Table 4, the "best-effort extractions from literature" are likely incorrect, at least for some cases.
E.g., for [52] ShallowConvNet and DeepConvNet:
* there had never been any evaluation on BCI2000. For BCIC IV Dataset 2a/2b reported metrics were accuracy and kappa, never F1. There was no LOSO or CV evaluation, but the results were evaluated within-subject following the official train-test split of those datasets. So it is entirely unclear how those results in the table were obtained

In general, there is also a confusion about BNCI and BCIC IV-2a/b.
BNCI 2014-001 is BCIC IV 2a and BNCI 2014-002 is BCIC IV 2b.
In A.1 the authors mention both of them in addition to BCI2000, in 4.1 the authors write about "**two** datasets [....] BCI 2000 [...] and  BNCI Horizon 2020 (004/008/009), with 9 subjects, 22 channels, and a sampling rate of 250 Hz" which would be consistent with BCIC IV 2a, however then it is unclear to me how exactly they compute F1 for multi class case? The table caption "Both data include left/right hand and foot MI." is again incosistent as none of BCIC IV 2a/b contain foot MI. The sentence "Other CNN variants (e.g., SCCNet, EEG-TCNet) have reported strong results on BCI Competition IV-2a, though their performance on BCI2000 or BNCI Horizon 2020 remains untested." does not make sense, again as BNCI 2014-001 is BCIC IV 2a and BNCI 2014-002 is BCIC IV 2b.

Also, while an ablation is helpful, the small differences (e.g.  with or without contrastive learning) suggest using multiple seeds to further check the statistical significance of the differences.

Overall, in my view, unfortunately these errors and inconsistenties question the entire reported results.

**Autoencoder spectrum effects**

Regarding the reported effects of the autoencoder on the spectrum:
"The DAE effectively suppresses high-frequency and non-physiological noise while preserving task-relevant spectral features, as demonstrated by the PSD plot."
Higher frequencies between 50-90 (high-gamma) can also be task-informative in some EEG datasets, so this statement is not true in general. Also it then would be interesting to see a comparison of a pure lowpass below 40 Hz to the autoencoder.

**Questions:**

See weaknesses

---

### Note · Authors · 2025-11-21

**Comment:**

Subject: Request to Make Withdrawn Submission Private / Remove Public Visibility
Dear Chairs,
I am the author of the submission:
Title: Contrastive and Multi-Task Learning on Noisy Brain Signals with Nonlinear Dynamical Signatures
Paper ID: 20504
I would like to withdraw the paper and keep it fully private (PDF, metadata, and reviews), but the OpenReview interface for this venue does not provide an option to control visibility during withdrawal.
Could you please make the withdrawn paper visible only to the authors and chairs, or alternatively remove its public entry?
Thank you for your help.
Best regards,
Sucheta

**Withdrawal Confirmation:**

I have read and agree with the venue's withdrawal policy on behalf of myself and my co-authors.